# Mechanobiological model for simulation of injured cartilage degradation via pro-inflammatory cytokines and mechanical stimulus

**Atte S. A. Eskelinen**[1]*, **Petri Tanska**[1], **Cristina Florea**[1,2], **Gustavo A. Orozco**[1], **Petro Julkunen**[1,3], **Alan J. Grodzinsky**[2], **Rami K. Korhonen**[1]

**1** Department of Applied Physics, University of Eastern Finland, Finland, **2** Departments of Biological Engineering, Electrical Engineering and Computer Science and Mechanical Engineering, Massachusetts Institute of Technology, Cambridge, United States of America, **3** Department of Clinical Neurophysiology, Kuopio University Hospital, Finland

* attees@uef.fi

**Data Availability Statement:** All the underlying data are available in Fairdata Research Data Storage Service (https://etsin.fairdata.fi/dataset/

## Abstract

Post-traumatic osteoarthritis (PTOA) is associated with cartilage degradation, ultimately leading to disability and decrease of quality of life. Two key mechanisms have been suggested to occur in PTOA: tissue inflammation and abnormal biomechanical loading. Both mechanisms have been suggested to result in loss of cartilage proteoglycans, the source of tissue fixed charge density (FCD). In order to predict the simultaneous effect of these degrading mechanisms on FCD content, a computational model has been developed. We simulated spatial and temporal changes of FCD content in injured cartilage using a novel finite element model that incorporates **(1)** diffusion of the pro-inflammatory cytokine interleukin-1 into tissue, and **(2)** the effect of excessive levels of shear strain near chondral defects during physiologically relevant loading. Cytokine-induced biochemical cartilage explant degradation occurs near the sides, top, and lesion, consistent with the literature. In turn, biomechanically-driven FCD loss is predicted near the lesion, in accordance with experimental findings: regions near lesions showed significantly more FCD depletion compared to regions away from lesions ($p<0.01$). Combined biochemical and biomechanical degradation is found near the free surfaces and especially near the lesion, and the corresponding bulk FCD loss agrees with experiments. We suggest that the presence of lesions plays a role in cytokine diffusion-driven degradation, and also predisposes cartilage for further biomechanical degradation. Models considering both these cartilage degradation pathways concomitantly are promising *in silico* tools for predicting disease progression, recognizing lesions at high risk, simulating treatments, and ultimately optimizing treatments to postpone the development of PTOA.

43c5e4f1-3c3e-4ea2-b63a-38aca480b77c). DOI:
https://doi.org/10.23729/0fec1760-d320-44f3-
baae-608aee509dc5

**Funding:** This work was supported by the Doctoral Programme in Science, Technology and Computing (SCITECO) of the University of Eastern Finland (http://www.uef.fi/en/web/dpsciteco; ASAE); the Finnish Cultural Foundation (https://skr.fi/en; grant number 00191044; PT); the Maire Lisko Foundation (PT); the Academy of Finland (https://www.aka.fi/en; grant numbers 286526 [RKK], 322423 [PJ], 324529 [RKK]); the Sigrid Jusélius Foundation (https://sigridjuselius.fi/en/; RKK); the Instrumentarium Science Foundation (ASAE); the European Union's Horizon 2020 research and innovation programme under the Marie Skłodowska-Curie grant agreement nos 702586, 713645 (https://ec.europa.eu/programmes/horizon2020/en/h2020-section/marie-sklodowska-curie-actions; RKK). The funders had no role in study design, data collection and analysis, decision to publish, or preparation of the manuscript.

**Competing interests:** The authors have declared that no competing interests exist.

## Author summary

Post-traumatic osteoarthritis is a musculoskeletal disorder where inflammatory processes and abnormal joint loading predispose articular cartilage to degradation after a mechanical injury. Since inflamed and injured cartilage cannot be reversed back to healthy state, prevention of osteoarthritis progression is advisable, a prestigious goal where computational models could serve as tools. The current literature is short of computational models combining both biochemical and biomechanical aspects of osteoarthritis. Thus, here we implemented inflammation of living cartilage tissue followed by biochemical perturbations of tissue homeostasis and shear strain-induced biomechanical degradation in novel cell-to-tissue-level finite element models. The models presented in this paper and enriched by our experimental findings/previous literature provide profound new mechanobiological insights and predictions about cartilage degradation in injured and inflamed tissue under physiologically relevant mechanical loading. We suggest that mechanobiological computational models could be applied as *in silico* analysis tools that provide clinicians information of the personalized progression of post-traumatic osteoarthritis and decision-making guidance for treatment of the disease.

## Introduction

Acute joint insult can debilitate the functioning of articular cartilage and lead to pain, joint stiffness and disability in patients. Moreover, trauma such as anterior cruciate ligament (ACL) rupture [1,2] may be associated with cartilage injury and increased susceptibility to cartilage degeneration which can culminate in post-traumatic osteoarthritis (PTOA) [3–5]. Two major mechanisms have been suggested to play a role in PTOA progression. The first mechanism involves joint inflammation [3,6,7] and the resulting diffusion of pro-inflammatory cytokines from the synovial fluid into cartilage [8–11], leading to proteolysis of cartilage matrix. The second mechanism involves biomechanical factors [12] including induction of chondral defects and aberrant loading characteristics in the knee joint milieu [13–15], leading to elevated shear strains near chondral lesions [16–18]. While these mechanisms have been recognized for years, current healthcare approaches lack effective tools to identify patients with increased risk of PTOA development. Prevention or even postponing the onset of the disease and surgical interventions would be desirable. Thus, we urgently need novel tools to predict disease progression.

Computational modeling is a critically important and cost-efficient approach which can serve as a predictive tool of possible PTOA progression. Finite element (FE) modeling has been successfully used to assess cartilage damage at both tissue [10,17,19] and joint levels [20,21]. Furthermore, subject-specific numerical models can be helpful in optimization of personalized intervention strategies, rehabilitation procedures, and surgeries. They could also be salutary in development of immunomodulating therapies [22,23] and disease-modifying osteoarthritis drugs [9,24,25]. When validated with experimental data, mechanobiological models can provide key quantitative and qualitative insights into biochemical and biomechanical adaptive processes in cartilage. However, computational models combining both inflammatory and loading mechanisms are scarce [26].

Inflammation plays a multifaceted role post-injury. The innate inflammatory response is an inherent part of host defense mechanisms aimed at maintaining tissue integrity and homeostasis. When tissue homeostasis is disturbed via inflammation, synovial macrophages [27,28] and elevated concentrations of inflammatory messenger molecules (such as alarmins, chemokines,

and cytokines) orchestrate signaling cascades that promote degradation of the extracellular matrix (ECM) of cartilage [3,29,30]. Cytokines originate primarily from cells in the synovial lining; subsequent increase in cytokine signaling can lead to recruitment of additional immune cells to the inflamed area, and the newly recruited cells produce even more cytokines. These small proteins diffuse into cartilage [8] and bind to cell receptors on chondrocyte (cartilage cell) surface, thus regulating the balance of anabolic and catabolic gene expression in cartilage [31]. Some cytokines in cartilage are pro-anabolic (anti-inflammatory), such as transforming growth factor beta (TGF-β), interleukin (IL)-1 receptor antagonist, IL-4, IL-10, and erythropoietin [22,28,32]. Other cytokines are pro-catabolic (pro-inflammatory), such as tumor necrosis factor alpha (TNFα), IL-1, IL-6, IL-8, IL-12, and IL-15 [3,8,23,33,34]. Previous experiments [4,35–37] suggest that chondrocytes affected by pro-inflammatory cytokines upregulate production of proteolytic aggrecanases ADAMTS-4,5 (a-disintegrin and metalloproteinase with thrombospondin motifs-4,5) and matrix metalloproteinases including the collagenases MMP-1,13. These proteases cleave matrix proteoglycans (PGs) and collagen fibrils, respectively. However, tissue inhibitors of metalloproteinases (TIMPs) can slow these catabolic processes by ADAMTS (specifically, TIMP-3 inhibits ADAMTS-4,5 [38]) and MMPs [39]. To predict the experimentally observed cartilage matrix loss [40], Kar and coworkers [9,10] developed a numerical model including the diffusion of IL-1 into cartilage and the subsequent degeneration of tissue aggrecan (the main PG in articular cartilage) and collagen. Other *in silico* studies [22,26,41] have also sought to cast new light on inflammation and complex biochemically-induced degradation mechanisms of cartilage.

Mechanical loading is also an essential regulator of cartilage homeostasis [42]. Previous experimental studies have proposed that the harsh loading environment of the knee joint [43], unfavorable loading frequency [44], and presence of chondral injuries [45] can predispose cartilage to biomechanical degradation. In efforts to replicate experimental findings, computational models have focused on the effects of mechanical loading-modulated tissue stresses (such as maximum principal stress) [20,46], strains [17,19,47], fiber stretches [41], volumetric deformations [48], interstitial fluid velocities [49], and altered tissue swelling properties [50] on induction of ECM degeneration. These models have included collagen fibril damage [41,46], PG or related fixed charge density (FCD) loss [17,49], tissue softening [47,51], and increase of tissue permeability [51].

Taking a step towards tools estimating PTOA progression, in this study we provide novel mechanobiological insights into biochemically and biomechanically-driven bovine cartilage degradation shortly after traumatic injury. Building on the previous research by Kar et al. [9,10] and Orozco et al. [49], we present a mechanobiological fibril-reinforced porohyperelastic swelling model of cartilage degeneration which considers biochemical (diffusion of pro-inflammatory IL-1 into tissue and subsequent release of ADAMTS) and biomechanical (elevated levels of the maximum shear strain during physiologically relevant dynamic loading) degradation mechanisms separately and simultaneously. We simulate spatial and temporal decreases in FCD and compare the numerical predictions with previous experimental results on cartilage plugs harvested from 1-2-week-old calves [35,36,40,49,52]. To the best of our knowledge, this is the first computational study involving the known extent of chondral lesions and including simultaneously both biochemical and biomechanical degradation pathways. We concentrate here on IL-1 due to its known role in cartilage inflammation [10,28] and the comparative lack of quantitative information on the dynamic temporal patterns of simultaneously acting inflammatory cytokines (such as IL-6, TNFα) and subsequent cartilage adaptation. Furthermore, we here include only FCD loss and not yet collagen degradation [10,41] or reorientation [53,54], since the reported experiments [49] were performed on a relatively short timescale (12 days); during this short time, substantial collagen degradation has not yet

occurred and thus collagen content was assumed to remain unaltered as suggested in earlier animal model experiments [55]. In addition, as one of the early changes in PTOA, FCD loss was presumed to occur prior to collagen damage [10,40,56], consistent with observations by Li et al. [40] and Kar et al. [10] that collagen loss was initiated only after 12 days of IL-1 culture.

See Table 1 for a list of abbreviations.

## Materials and methods

### Experiments

Previous experimental research has assessed cytokine-mediated and biomechanically-driven cartilage degeneration [36,40,49,52,57,58]. These studies involved evaluation of **(1)** loss of sulphated glycosaminoglycans (GAGs) to culture medium via the dimethylmethylene blue (DMMB) assay, or **(2)** loss of FCD via decrease in optical density obtained with digital densitometry (DD) of Safranin-O stained sections [59,60]. Both methods are indicative of PG matrix damage. In the current study, we quantified average FCD loss near and further away from cartilage lesions based on the data of Orozco et al. [49] to **(1)** characterize how the presence of lesions affects early matrix damage and PG release, and **(2)** to compare biomechanically-induced FCD loss with our computational estimates. Due to lack of data in the literature, we did not focus on localized FCD loss in injured samples cultured with cytokines. See S1 Supplementary Material (Subsections S1.1–1.4) for additional details on the previously reported experimental data used for the present modeling analyses.

### Computational mechanobiological models

We developed mechanobiological models to predict the loss of cartilage FCD content over time caused by inflammatory and shear strain mechanisms. In the biochemical model (Fig 1A), FCD loss was associated with diffusion of the cytokine IL-1 into cartilage and subsequent

**Table 1. List of abbreviations.**

| Abbreviation | Description |
| --- | --- |
| 2D/3D | 2-dimensional/3-dimensional |
| ACL | anterior cruciate ligament |
| ADAMTS | a-disintegrin and metalloproteinase with thrombospondin motifs |
| DD | digital densitometry |
| DMMB | dimethylmethylene blue |
| ECM | extracellular matrix |
| FCD | fixed charge density |
| FE | finite element |
| FRPHES | fibril-reinforced porohyperelastic swelling |
| GAG | glycosaminoglycan |
| IL | interleukin |
| MMP | matrix metalloproteinase |
| OD | optical density |
| PG | proteoglycan |
| PTOA | post-traumatic osteoarthritis |
| sIL-6R | interleukin-6 soluble receptor |
| TGF-β | transforming growth factor beta |
| TIMP | tissue inhibitors of metalloproteinases |
| TNFα | tumor necrosis factor alpha |

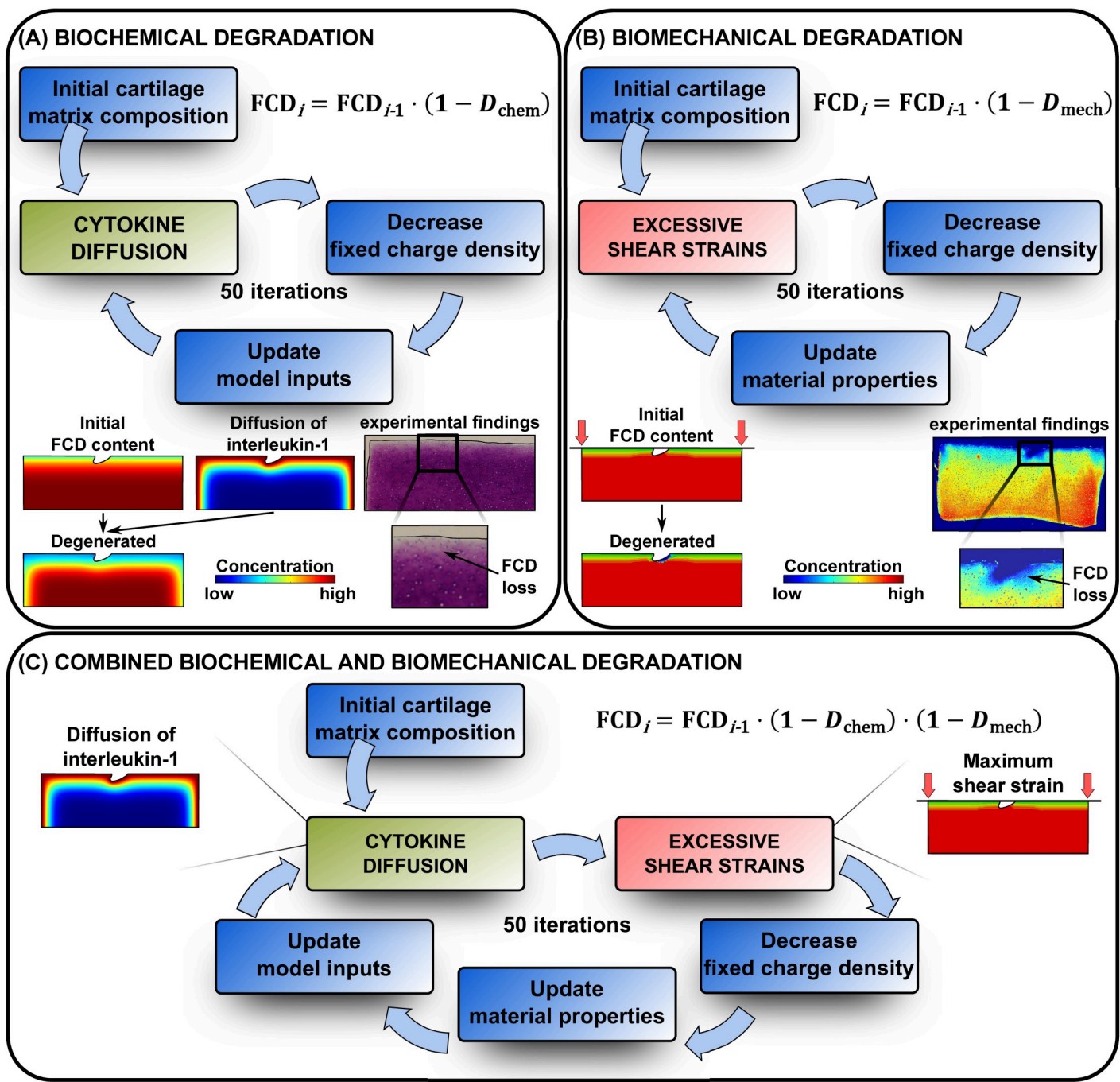

**Fig 1. Cartilage degeneration algorithm.** Outline of the proposed proteoglycan degeneration algorithm based on the loss of fixed charge density (FCD) via A) diffusion of pro-inflammatory cytokine interleukin-1 into the tissue and subsequent biochemical degradation of cartilage matrix, B) excessive levels of the maximum shear strain especially near lesions and C) combination of these cartilage-degrading mechanisms. In the combined degradation model, both mechanisms are in effect simultaneously during each simulation iteration, followed by an update of non-fibrillar matrix contents for the next iteration. Experimental biomechanical degradation images modified with permission from Orozco et al. [49].

increased levels of ADAMTS proteases. In the biomechanical model (Fig 1B), the decrease in the FCD content was driven by excessive levels of the maximum shear strain. Lastly, we implemented both of these mechanisms simultaneously in a combined biochemical and

biomechanical model (Fig 1C). The finite element (FE) model simulations were conducted using COMSOL Multiphysics (version 5.3a, Burlington, MA, USA) for the biochemical model, ABAQUS (2018, Dassault Systèmes, Providence, RI, USA) for the biomechanical model, and both software packages simultaneously for the combined model. Furthermore, a custom-made interface was developed in MATLAB (R2017b, The MathWorks, Inc., Natick, MA, USA) for handling the flow of PG matrix degradation data between COMSOL and ABAQUS. The cartilage disk thickness and radius for the 2D FE model were $h = 1$ mm and $r = 1.5$ mm, respectively. The cartilage lesion geometry (depth 144 μm, width 316 μm) was adopted from the study by Orozco et al. [49] in which the FE mesh generated in ABAQUS was based on segmentation of a histological slice from a representative injured cartilage disk. The output of all the models was the regional average FCD loss (decrease in FCD levels with respect to initial levels) over time for the regions **(1)** within 0.1 mm (±10%) from the lesion, and **(2)** 0.15 mm deep (±10%, from the surface) and 0.45 mm wide (±10%) away from the lesion at a location in the middle between the lesion and disk edges, as performed in the analysis of experimental biomechanical FCD depletion (see S1 Supplementary Material Subsection S1.2). Moreover, we created an intact reference model to analyze the FCD loss from the middle of the disk in a same-sized region as away from the lesion in the injury model. In all the models, the possible minimum FCD concentration was set to 10% of the initial minimum FCD value (found from the superficial zone) to avoid convergence issues. The most essential model inputs, outputs, and equations defining the computational FCD loss are recapitulated in Fig 2. For more details, readers are referred to Kar et al. [10], Orozco et al. [49], and the S1 Supplementary Material (Subsections S1.5–1.9, S1 Table).

## Simulation of biochemical degradation

Biochemically-driven aggrecan loss was modeled via diffusion of IL-1 cytokines (1 ng/ml in the culture medium) into tissue and subsequent increase in protease biomolecule concentrations (time integration in COMSOL Multiphysics). In short, the model [10] considered the net degradative effect of ADAMTS-4,5 and TIMP-3 on aggrecan content. The model inputs include effective diffusion coefficients of IL-1 [10,61], intact aggrecan [62–64], and aggrecanases [65], aggrecan biosynthesis rate depending on the local IL-1 concentration [10,66–68], initial aggrecan distribution [69,70], catabolic rate for aggrecan cleavage by aggrecanases [71], and production/degradation of ADAMTS [10,72]. In the COMSOL model, the primary variable of interest was intact aggrecan concentration, the changes of which were interpreted as changes in FCD concentration in ABAQUS (see Subsection Materials and methods: Biochemical fixed charge density loss; ultimately, the biochemical model output was FCD loss). We chose to model biochemical degradation for 21 days because in the model by Kar et al. [10] the basal rate of aggrecan synthesis and the mass transfer coefficient of aggrecan were calibrated within this timeframe with respect to PG loss in young animals [66]. In COMSOL, we selected four-node linear quadrilateral plate elements (element type CQUAD4) with the same node locations as in the ABAQUS mesh. This mesh was imported into COMSOL in a NASTRAN-format file generated with a custom-made MATLAB script.

## Cytokine diffusion model

The diffusion of IL-1 into porous biphasic cartilage and proteolytic effects of aggrecanases were modeled with parabolic reaction—diffusion partial differential equations [10,73]

$$\frac{\partial C_j}{\partial t} = D_j \nabla^2 C_j \pm R_j, \tag{1}$$

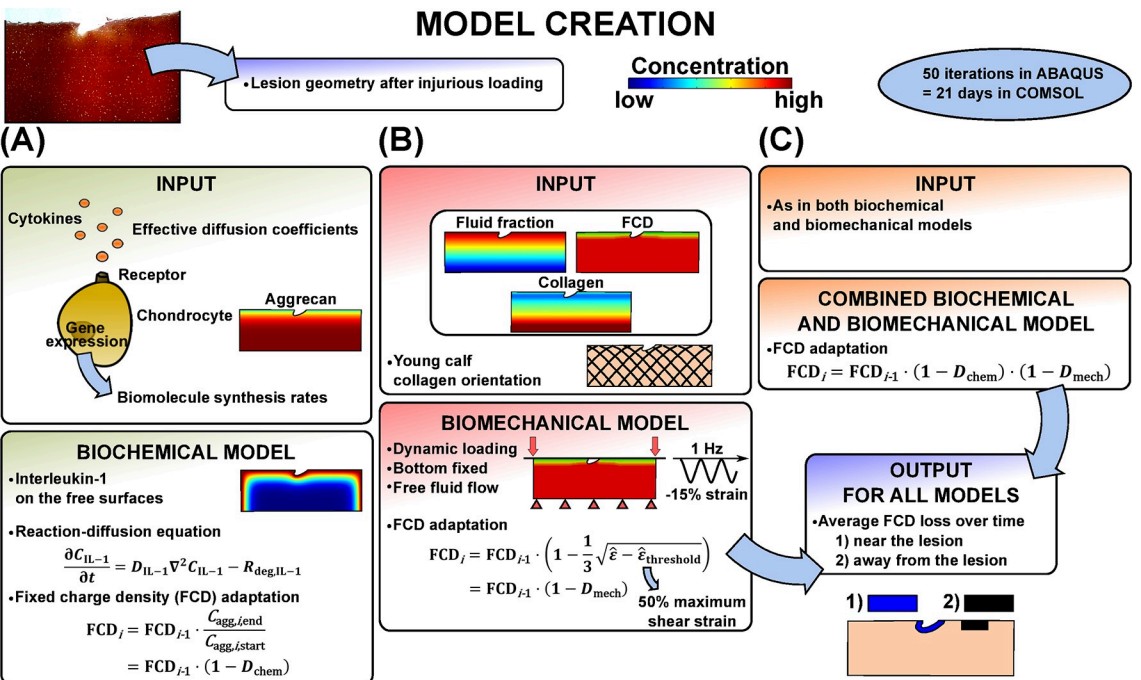

**Fig 2. Workflow.** Workflow for creation of A) biochemical, B) biomechanical and C) combined biochemical and biomechanical degradation models. Lesion geometry was based on histological findings after injurious loading. All of the mechanobiological models estimated fixed charge density (FCD) loss near and away from the lesion over time. Histology image obtained with permission from Orozco et al. [49].

where $C_j$ is the concentration/amount of constituent $j$, $t$ is time, $D_j$ is the effective diffusivity of chemical species $j$ and $R_j$ is the corresponding source/sink term, which describes the rate of generation/repair or degradation/consumption of individual species. The species $j$ includes chondrocytes, IL-1, intact aggrecans, and aggrecanases. The source/sink terms $R_j$ for proteolytic aggrecanases were designed in accordance with Michaelis—Menten kinetics. Experimentally observed time delay in secretion of ADAMTS after initiation of biochemical challenge [74] was included in the form of a stimulus equation [10]

$$\frac{\partial S_1}{\partial t} = \alpha_1(C^* - S_1),\tag{2}$$

where $S_1$ is a variable describing IL-1-mediated stimulus response of ADAMTS, $C^*$ is the concentration of IL-1-IL-1-receptor complexes and $\alpha_1$ is rate constant for stimulus. This stimulus variable $S_1$ was used to define source terms $R_j$ in Eq (1) for ADAMTS.

## Initial and boundary conditions in biochemical model

As initial condition, the concentration of IL-1 and aggrecanases were set to 0 ng/ml, and aggrecanase-stimulus ($S_1$ in Eq (2) at time $t = 0$ d) to 0 mol/m$^3$ inside the tissue. The chondrocyte concentration [75] was uniform and constant throughout the simulation, and no apoptosis nor necrosis was modeled due to lack of cell death rate (dead cells/(s·m$^3$)) data in the literature.

For the IL-1 concentration, Dirichlet boundary condition of 1 ng/ml was set on the free surfaces (top, lesion, and lateral) throughout the simulation. We assumed a constant cytokine level outside tissue since during the experiments the culture medium was changed every other

day. For the same reason, concentration of aggrecanases was set to 0 ng/ml on the free and bottom surfaces. Mass transfer of intact aggrecan was allowed through the free surfaces. This was handled by Robin boundary conditions (combination of Dirichlet and Neumann boundary conditions) to account for diminished diffusion of chemical species through the cartilage-culture medium interface [76]. No mass transfer nor diffusion was allowed for any of the chemical species through the bottom surface.

## Biochemical fixed charge density loss

In the COMSOL model, the primary degrading constituent is intact aggrecan, having concentration $C_{agg}$. Intact aggrecan concentration is affected by the effects of aggrecanases and by synthesis of new aggrecans by the chondrocytes. However, to compare the FCD concentration distribution (which is the primary variably of interest in ABAQUS model) between all the mechanobiological models, intact aggrecan concentration in COMSOL was converted to FCD concentration in ABAQUS. Therefore, relative changes in integration point-wise aggrecan concentrations during one simulation iteration $i$ (50 iterations in total corresponded to 21 days of simulated time, so each iteration simulated a time period of $21/50$ d $\approx 10$ h; see more details from Subsection Materials and methods: Simulation of combined biochemical and biomechanical degradation) were implemented with MATLAB as coefficients of FCD depletion for each integration point

$$\text{FCD}_i = \text{FCD}_{i-1} \cdot \frac{C_{agg,i,end}}{C_{agg,i,start}} = \text{FCD}_{i-1} \cdot (1 - D_{chem}), \qquad i = 1, 2, \ldots, 50 \qquad (3)$$

where $\text{FCD}_i$ is the integration point-wise FCD concentration in ABAQUS at iteration $i$, $C_{agg,i,end}$ and $C_{agg,i,start}$ are the integration point-wise aggrecan concentrations in COMSOL at the end and start of iteration $i$, and $D_{chem} = \left(1 - \frac{C_{agg,i,end}}{C_{agg,i,start}}\right)$ is the biochemical degradation rate. The mesh was the same in both COMSOL and ABAQUS. After each iteration, the decreased FCD in each integration point was imported to COMSOL as new integration point-wise aggrecan concentration for the following iteration $C_{agg,i+1,start}$

$$C_{agg,i+1,start} = C_{agg,i,start} \cdot \frac{\text{FCD}_{i-1}}{\text{FCD}_i}, \qquad i = 1, 2, \ldots, 50 \qquad (4)$$

where for the biochemical model $C_{agg,i+1,start} = C_{agg,i,end}$. This was done with the COMSOL import tool, which used linear interpolation (see S1 Supplementary Material Subsection S1.6).

## Simulation of biomechanical degradation

Biomechanical degradation of PG content was associated with increased levels of intra-tissue shear strain during loading, since shear strains have been suggested to increase near chondral lesions [16–18,77] and may induce cartilage matrix breakdown [19] and chondrocyte apoptosis [78–80]. Ultimately, this leads to loss of FCD content. The biomechanical model solved a highly non-linear problem consecutively (50 times, updating FCD content between iterations) with a transient SOILS analysis in ABAQUS to account for the viscoelastic behavior of the fibril-reinforced porohyperelastic material with Donnan osmotic swelling and chemical expansion (see S1 Supplementary Material Subsection S1.7 for further details on the material model). The structural inputs for the biomechanical model included depth-dependent fluid fraction [81], FCD [49], and collagen distributions [82], and collagen fibril orientation in immature tissue [49,83]. The output of the biomechanical model was FCD loss. Linear axisymmetric continuum elements with pore pressure (element type CPE4P) were implemented for the model in ABAQUS.

## Boundary conditions in biomechanical model

Initially, the cartilage was allowed to reach mechanical equilibrium with external salt concentration via swelling [84]. Then the cartilage disk was subjected to the same dynamic loading protocol (modeled for two full loading-unloading cycles (two seconds), since our preliminary testing suggested that a longer simulation did not cause any substantial changes in strains) with an impermeable platen as in the experiments. During loading, the cartilage-platen contact was frictionless (contact type: *surface-to-surface*; contact behavior: *penalty*), the explant was allowed to bulge radially (bottom surface was fixed axially, one bottom corner node was also fixed horizontally to prevent the disk from rotating and translating), and fluid flow was allowed through the free surfaces.

## Biomechanical fixed charge density loss

The FCD content was decreased in each integration point when the maximum shear strain levels [17,85]

$$\hat{\varepsilon} = \max\{|\varepsilon_{p,1} - \varepsilon_{p,2}|, |\varepsilon_{p,1} - \varepsilon_{p,3}|, |\varepsilon_{p,2} - \varepsilon_{p,3}|\},\tag{5}$$

exceeded the degeneration threshold of $\hat{\varepsilon}_{threshold} = 50\%$, where $\varepsilon_{p,k}$ are the principal strains of the Green—Lagrangian strain tensor. This threshold value was chosen to match experimental findings and predicted biomechanical FCD loss as done by Orozco et al. [49]. The integration point-wise maximum shear strain $\hat{\varepsilon}$ during loading (at the second peak of compressive 1-Hz load during two-second simulation) and the degeneration threshold $\hat{\varepsilon}_0$ defined together the biomechanical degeneration rate $D_{mech}$ [17,51]

$$D_{mech} = \frac{1}{3}\sqrt{\hat{\varepsilon} - \hat{\varepsilon}_{threshold}}.\tag{6}$$

See also the Subsection Materials and methods: Simulation of combined biochemical and biomechanical degradation for justification of this equation. Thus, the integration point-wise FCD loss for simulation iteration $i$ was defined as

$$FCD_i = FCD_{i-1} \cdot (1 - D_{mech}), \qquad i = 1, 2, \ldots, 50.\tag{7}$$

In total, the model was run for 50 iterations to reach near-equilibrium (see Subsection Results: Simulation of biomechanical degradation) in terms of the predicted FCD distribution. The loss of FCD affected the Donnan osmotic swelling pressure gradient $\Delta\pi$, chemical expansion stress $T_c$, and ultimately the total stress tensor $\sigma_{tot}$ (see S1 Supplementary Material Subsection S1.7 for definition of these terms). After each iteration, the decreased FCD was imported as a new FCD distribution for the following ABAQUS simulation iteration ($c_{F,0}$ in Eq. (S15) in S1 Supplementary Material Subsection S1.7).

## Simulation of combined biochemical and biomechanical degradation

In a third model, both biochemical and biomechanical degradative mechanisms were acting concomitantly. (See Subsections Materials and methods: Initial and boundary conditions in biochemical model and Materials and methods: Boundary conditions in biomechanical model above for initial and boundary conditions for both of these mechanisms, respectively.) The combined FCD loss was implemented similarly as in Eqs (3) and (7)

$$FCD_i = FCD_{i-1} \cdot (1 - D_{chem}) \cdot (1 - D_{mech}), \qquad i = 1, 2, \ldots, 50\tag{8}$$

and the integration point-wise FCD loss in ABAQUS was used to solve the intact aggrecan

distribution to be imported to COMSOL with Eq (4). Similar approach (multiplication of the biochemical and biomechanical degradation effects) has also been previously used [26].

Simulation time of 21 days in COMSOL was chosen to correspond to 50 iterations in ABA-QUS. 21 days was used originally for the biochemical model calibration [10] and 50 iterations was found to result in near-equilibrium of biomechanically-driven FCD loss (negligible change in FCD loss with more iterations, see Subsection Results: Simulation of biomechanical degradation). To further justify our decision, the biomechanical degeneration rate (Eq 6) [17,51] combined with this time scale and number of iterations captures well the FCD loss near lesions in our experiments (see Subsection Results: Simulation of biomechanical degradation and S1 Supplementary Material Subsection S2.1) and the rapid early loading-associated GAG loss in mechanically loaded samples [83,86–88] (see Subsection Discussion: Biomechanical degradation).

### Mesh density

The sensitivity of the predicted FCD loss on the FE mesh was tested with five different meshes and the algorithm including both biochemically and biomechanically-driven degradations. These meshes were increasingly dense near the lesion and the free edges. A mesh with 918 elements was chosen, since increasing the mesh density beyond that resulted in only minor changes in predicted FCD loss within 0.1 mm from the lesion and away from the lesion (S1 Supplementary Material Subsection S1.9, S1 Fig).

## Results

### Summary of cartilage degradation

Representative samples of experimental biochemically-driven degradation (Fig 3A) demonstrated progressive FCD loss starting from the free surfaces and extending over time to the deeper tissues. Mechanically injured and dynamically loaded samples showed pronounced FCD loss near defects (Fig 3B). Simulated biochemical degradation occurred near the top surface, edges and below the lesion (Fig 4). On the other hand, with biomechanical degradation algorithm we predicted matrix losses only near the lesion (Fig 5). In the model combining both mechanisms, matrix losses occurred near the free surfaces and especially in the vicinity of the lesion (Fig 6).

### Simulation of biochemical degradation

Biochemical degradation occurred near the free surfaces in both intact reference (Fig 4A) and injury (Fig 4B) models. At timepoint $t = 4$ d, the predicted average FCD losses were 38% for the intact reference model, and 44% both near and away from the lesion for the injury model. Below the lesion, the ECM was degraded deeper in the tissue than closer to the sides. Near the edge of the lesion in the superficial zone, the maximum FCD losses were 58% ($t = 4$ d) and 97% ($t = 12$ d) (Figs 4B and 7A). Temporally, the average regional biochemical FCD loss was of sigmoidal shape; first occurring at a relatively slow pace, then attaining the highest rate of degradation at around day 4, and finally reaching the maximum loss set up by our algorithm (10% of the initial minimum FCD was the lowest allowable FCD content) at around day 8 (Fig 7A). The localized peak FCD loss profile with logarithmic-like shape showed a rapid degradation near the edge of the lesion (Fig 7A). However, the overall degradation of cartilage did not cease at day 8 as shown by a steady increase of the degraded area after day 8 (Fig 8A; cartilage was referred as "degraded" if element-wise FCD loss compared to initial FCD content was

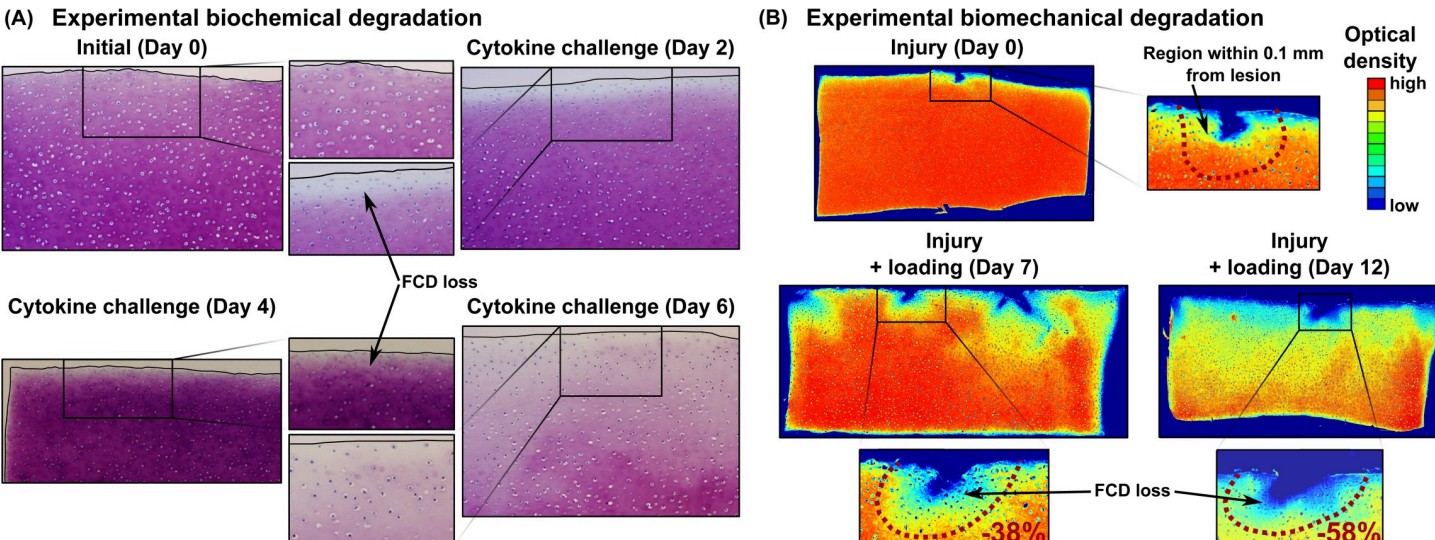

**Fig 3. Previous experimental findings.** A) *In vitro* experiments with exogenous pro-inflammatory cytokine interleukin-1 (IL-1) challenge (10 ng/ml in culture medium) show marked matrix degradation near the sample surface and edges [40]. B) *In vitro* experiments with injurious loading (50% at rate 100%/s) following a dynamic loading period (15% strain amplitude, haversine waveform, 1 Hz, 1h on and 5h off cycles) in unconfined compression show lower optical density (~lower fixed charge density, FCD) especially near lesions. The numbers [%] show the average localized FCD loss compared to the day when cartilage was mechanically injured. Biomechanical degradation images modified with permission from Orozco et al. [49].

equal or more than 20%). The degraded area increased rapidly around day 2 (Fig 8A) starting from the free surfaces (see S1 Supplementary Material, Subsection S2.4, S5 Fig, S1 Animation).

## Simulation of biomechanical degradation

The intact reference model showed no FCD loss anywhere (Fig 5A), whereas the injury model showed an average FCD loss of 35% near the lesion at timepoint $t = 12$ d (Fig 5B). The localized peak FCD losses followed logarithmic-like shape and were 89% ($t = 4$ d) and 97% ($t = 12$

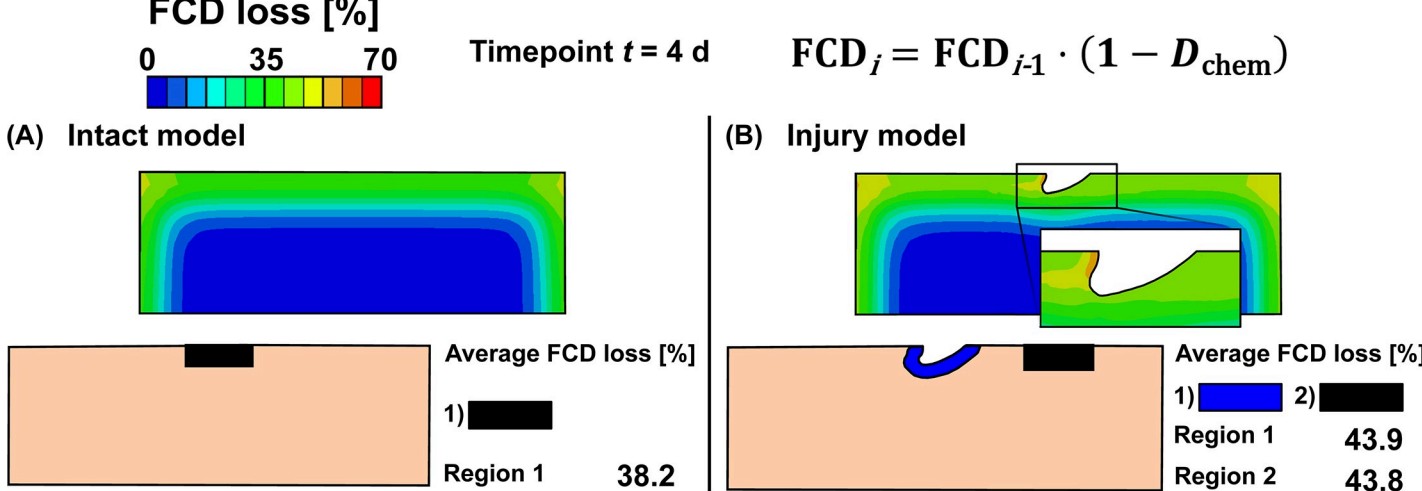

**Fig 4. Simulated biochemical cartilage degradation.** Predicted biochemical fixed charge density (FCD) losses in A) intact reference model and B) injury model. Biochemical degradation shown at time $t = 4$ d was simulated with 1 ng/ml of exogenous interleukin-1.

## Simulated biomechanical degradation

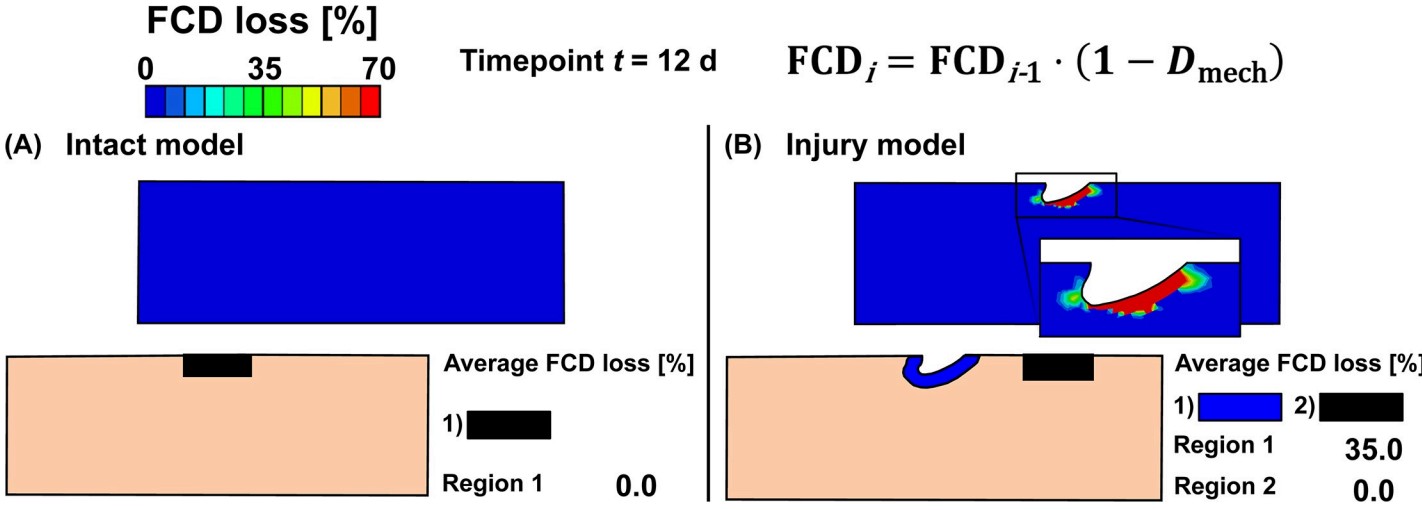

**Fig 5. Simulated biomechanical cartilage degradation.** Predicted biomechanical fixed charge density (FCD) losses in A) intact reference model and B) injury model. Biomechanical degradation shown at time $t = 12$ d was simulated with the degeneration threshold of $\hat{\varepsilon}_{threshold} = 50\%$ of the maximum shear strain. This threshold was chosen to match model predictions and experimental findings.

d) right below the lesion surface (Figs 5B and 7B). The average regional FCD loss occurred at a fast rate early-on (days 0–5), then leveling off to a slower rate over time (Fig 7B). At day 21, the predicted FCD distribution had reached near-equilibrium (Fig 7B, solid blue line; the FCD loss changed less than 0.15% per iteration after iteration 40). An addition of the base degeneration rate term (indicating spontaneous GAG loss from unloaded samples via passive diffusion, see S1 Supplementary Material Subsection S1.8) increased the predicted average FCD loss (Fig 7B). Then, the model predictions suggested a 6-7-fold (at day 2) and a 3-fold (at day 12)

## Simulated combined biochemical and biomechanical degradation

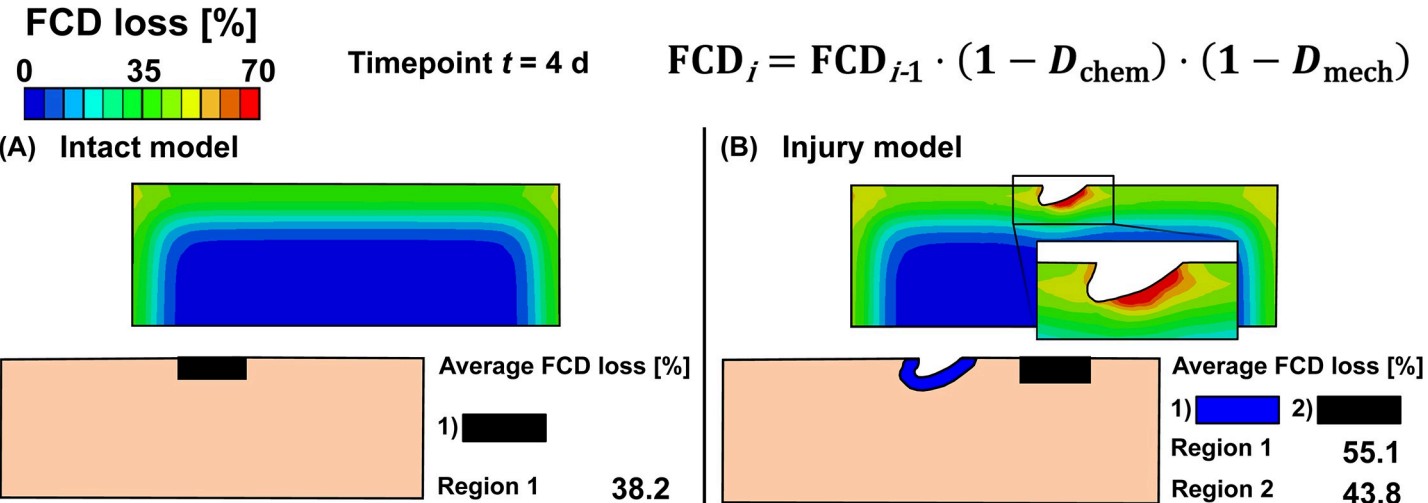

**Fig 6. Simulated combined biochemical and biomechanical cartilage degradation.** Predicted combined biochemical and biomechanical fixed charge density (FCD) losses in A) intact reference model and B) injury model. Combined biochemical and biomechanical degradation shown at time $t = 4$ d was simulated with 1 ng/ml of exogenous interleukin-1 and with the degeneration threshold of $\hat{\varepsilon}_{threshold} = 50\%$ of the maximum shear strain. This threshold was chosen to match model predictions and experimental findings.

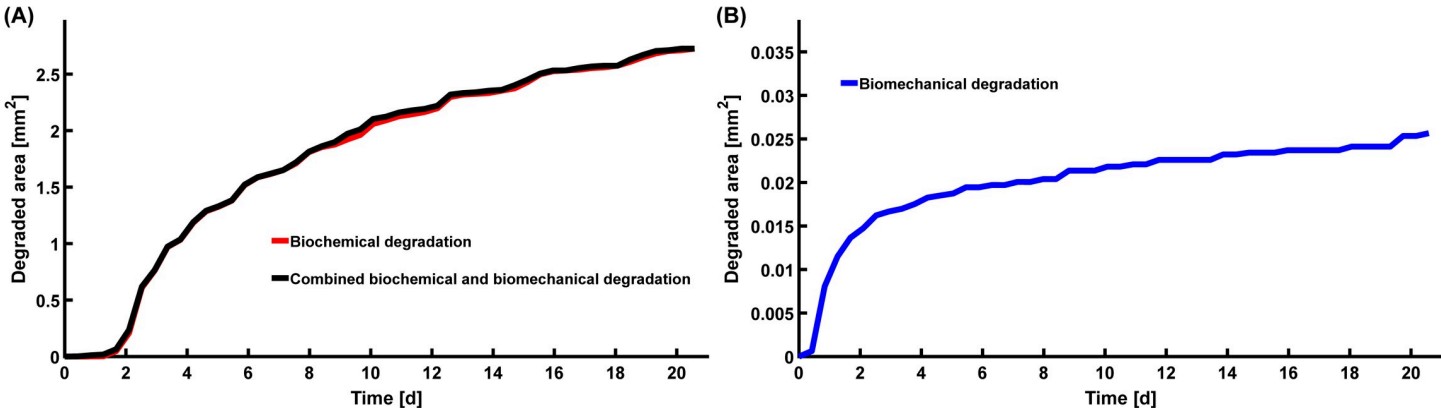

**Fig 7. Temporal estimations of cartilage degradation.** Average and maximum fixed charge density (FCD) losses over time predicted by A) biochemical, B) biomechanical, and C) combined biochemical and biomechanical model. The dashed line in biomechanical model includes the base degeneration rate parameter, which was chosen according to the experimental FCD loss in freely-swollen cartilage disks (see S1 Supplementary Material, Subsection S1.8). The blue and black errorbars (with 95% confidence intervals) at days 7 and 12 represent the experimental average FCD loss near lesions after dynamic loading and in freely-swollen samples, respectively. The FCD concentration was not allowed to decrease below 10% of the initial minimum FCD concentration in the model due to computational stability.

**Fig 8. Simulation of degraded cartilage area.** Temporal estimations of degraded cartilage area with A) biochemical and combined biochemical and biomechanical degradation and B) biomechanical degradation. In these simulations, cartilage was considered"degraded" when FCD loss in elements was equal or greater than 20%. See S1 Supplementary Material (Subsection S2.4, S5 Fig, S1–S4 Animation) for animations showing the propagation of the degraded area over time.

increase in FCD loss in areas near the lesion compared to areas away from the lesion (where base degeneration accounts for all of the FCD loss; Fig 7B). The degraded area increased early (days 0–2) at a fast pace and slowly throughout the rest of the simulation (Fig 8B and S1 Supplementary Material Subsection S2.4, S5 Fig, S2 Animation and S3 Animation).

### Simulation of combined biochemical and biomechanical degradation

The intact reference model predicted only biochemical degradation (38% FCD loss at $t = 4$ d, Fig 6A). However, combined biochemical and biomechanical degradation occurred near the free surfaces and especially in the vicinity of the lesion in the injury model (Fig 6B). The predicted average FCD losses were 55% and 44% near and away from the lesion, respectively. The localized maximum FCD losses were 81% ($t = 4$ d) and 98% ($t = 12$ d) below the lesion surface (Figs 6B and 7C). The FCD loss reached deeper layers of cartilage below the lesion compared to the tissues near lateral edges. During the early days of simulation (days 0–6), the predicted FCD loss near the defect was higher than away from the defect, where the FCD loss profile exhibited sigmoidal behavior (Fig 7C). Furthermore, the rate of FCD loss was almost constant near the lesion during the early days and initially higher near the lesion than away from it (Fig 7C). Below the lesion, the logarithmic-shaped maximum FCD loss profile predicted fast degradation (Fig 7C). At around day 8, the average FCD loss near and away the lesion reached equilibrium, but the overall cartilage degeneration continued (Fig 8A). Animations in the S1 Supplementary Material (Subsection S2.4, S5 Fig, S4 Animation) and Fig 8A show the rapid increase of degraded areas near the lesion and subsequently near the free surfaces.

## Discussion

### Summary

In this study, we have developed a novel mechanobiological FE model capable to predict cartilage degradation via both biochemically and biomechanically-driven mechanisms. We provided spatio-temporal estimations of matrix damage, which are in accordance with our experimental findings (S1 Supplementary Material Subsection S2.1, S2 Fig) and the current literature reports. With the chosen concentration of pro-inflammatory cytokine IL-1 (1 ng/ml, representing moderate inflammation), the ECM was degraded more dramatically over larger areas (near free surfaces and below the lesion) compared to the effect of mechanical loading alone after injury. The shear strain-modulated biomechanical degeneration that occurred only near the lesion was rapid, especially during 1–3 days following injury.

### Biochemical degradation

The intact reference model predicted slightly smaller FCD loss (38%) than the injury model both near (44%) and away from (44%) the lesion (Fig 4). This might be associated with the increased surface area available (lesion surfaces) for cytokines to diffuse into the tissue in the injury model. Likewise, this geometrical alteration may explain the increase in FCD loss near the edge of the lesion (peak FCD loss 58%) and just below the lesion compared to regions away from the defect. Therefore, the mere presence of lesions might affect biochemical degradation to some degree.

For comparison of biochemical model and experiments, Kar et al. [10] already provided data showing sound agreement between their model and experimental IL-1-induced degradation of young calf cartilage in terms of time-dependent aggrecan loss to the culture medium [40]. In addition, Patwari et al. [35] reported an average total GAG loss of 280 μg/ml per disk (600% more compared to untreated controls) over 6-day treatment using 10 ng/mL IL-1, *i.e.*,

10-fold higher cytokine concentration compared to that used in our model. Similarly, Stevens et al. [36] found that cartilage disks treated with 10 ng/ml of IL-1 for five days released aggrecan 2.6 times the amount released by control cartilage disks. These findings are supportive of the predictions of our biochemical model using 1 ng/ml of IL-1. FCD losses in bulk of the tissue (see S1 Supplementary Material Subsection S2.2, S3 Fig) were 31% (day 5) and 39% (day 6), while near the lesion they were 56% (day 5) and 67% (day 6). Furthermore, when we decreased the IL-1 levels to physiological levels in synovial fluid (0.01 ng/ml [89]) the predicted FCD loss decreased drastically (at day 4, -70% near the lesion and -83% away from the lesion) compared to treatment with 1 ng/ml of IL-1 (S1 Supplementary Material Subsection S2.3, S4 Fig).

## Biomechanical degradation

Biomechanical degradation models predicted localized loss of FCD near the lesion. These predictions were in good agreement with our experimental findings; loss of FCD near lesions was significantly higher than further away from the lesions (S1 Supplementary Material Subsection S2.1, S2 Fig). These experimental data and the biomechanical degradation model together suggest that the presence of lesions themselves is the primary reason for biomechanically-induced loss of FCD since dynamic loading itself did not degrade the ECM of intact cartilage. In support of these model simulations, we found experimentally that significantly more FCD was lost locally near lesions than away from them, while the average FCD concentrations were similar **(1)** away from lesions, **(2)** in freely-swollen cartilage disks, and **(3)** in uninjured dynamically loaded disks (see S1 Supplementary Material Subsection 2.1, S2 Fig). In addition, inclusion of the base degeneration rate (spontaneous degeneration of free swelling samples) led to a better fit with the experimental findings compared to the original injury model (Fig 7B).

Research quantifying the loss of GAG over time associated with the combination of initial impact injury followed by physiologically relevant dynamic loading are scarce [52] and, to the best of our knowledge, the current literature lacks data of localized GAG loss under aforementioned conditions. In contrast, studies using injurious loading alone [83,87,90] report rapid early GAG depletion. Specifically, Thibault et al. [86] found significantly higher GAG release rates during the first four days after loading compared to unloaded controls. Additionally, Rolauffs et al. [83] reported rapid GAG loss (~20%) by 48h after injurious loading (using the same injury-loading protocol as in the current study), comparable to our predictions obtained with biomechanical degradation rate $D_{mech}$ in Eq (6). Similarly fast loss of GAG was reported by DiMicco et al. [87] and Mohanraj et al. [88] in mechanically injured samples. Mohanraj et al. found early damage in cartilage tissue analog constructs after already 12, 24, 48, and 120 hours after injury (50% or 75% strain amplitude with rate 50%/s). Furthermore, they found a 2-3-fold increase in bulk PG matrix loss in injured constructs compared to uninjured controls by 48h after injury. This was predicted quite well by our biomechanical degradation model with base degeneration at $t = 2$ d (6-7-fold increase in localized FCD loss near the lesion compared to away from the lesion, see Subsection Results: Simulation of biomechanical degradation, and 2.2-fold increase in bulk FCD loss in the injury model versus intact reference model, see S1 Supplementary Material Subsection S2.2, S3 Fig).

## Combined biochemical and biomechanical degradation

The intact reference model combining both biomechanical and biochemical degradation mechanisms exhibited the same FCD loss as the biochemical model (38%, Figs 4A and 6A). The injury model revealed marked depletion of FCD near the lesion compared to regions away from the lesion. Moreover, the FCD loss predicted by the combined degradation model

and that by the biochemical model was essentially identical away from the defect. This suggests that biomechanical degradation does not play a marked role away from lesions even in the case when those regions are already affected by biochemical perturbations.

The maximum loss of FCD at the lesion site predicted by the combined biochemically and biomechanically-driven model was slightly smaller compared to model driven only by bio-mechanical degradation mechanism (for example, at timepoint $t$ = 4 d the combined and bio-mechanical models predicted 81% and 89% of peak FCD loss, respectively, see Fig 7C). The reason for this is that the combined model considers both passive diffusion of aggrecan and synthesis of new aggrecan, and their combined effect decreases the net rate of aggrecan loss. In contrast, the biomechanical model does not include aggrecan synthesis and, thus, does not include mechanisms for generation of new FCD.

Experimental literature that includes the known extent of quantified cartilage cracks along with biochemical degradation (*e.g.*, IL-1-induced) and dynamic loading is lacking. However, Li et al. [52] reported 40% total GAG loss after an 8-day treatment with 10% strain amplitude dynamic loading in the combined presence of IL-6, IL-6 soluble receptor (sIL-6R) and TNFα. Similarly, Sui et al. [34] reported 45% total GAG loss after a 6-day treatment with these same inflammatory cytokines and injurious loading just prior to the cytokine culture. These results are in accordance with our numerical predictions in bulk of the tissue (see S1 Supplementary Material Subsection S2.2, S3 Fig) with only 1 ng/ml of IL-1 after 6 days (42% FCD loss) and 8 days (61%) of simulation. Furthermore, the predicted average FCD losses near the lesion were 77% (day 6) and 95% (day 8) (Fig 7C).

## Limitations

Several limitations in this study are worth mentioning. Characterizing the temporal dynamics associated with combinations of inflammatory cytokines following mechanical injury is a difficult task. Due to lack of quantitative data about the interplay of different cytokines, we modeled only the net effect of IL-1, ADAMTS-4,5, and TIMP-3. In the end, a model including the whole range of pro- and anti-inflammatory cytokines with their individual activation characteristics would most likely be computationally very demanding. However, such a model might not predict substantially different spatio-temporal matrix degradation compared to the models used in this study. Lack of quantitative experimental data also makes it difficult to include cytokine-mechanical loading-interactions into our models. Although some evidence exists that injurious loading stimulates a few selected signaling pathways also stimulated by IL-1 and TNFα [29,91], modeling the dynamics of these pathways and their subsequent effect on cartilage composition was out of scope of this study. Also, the predictions of combined effect of cytokines and mechanical injury on the localized FCD loss especially at the lesion site were not compared to experimental findings due to lack of data in the literature.

Several model parameters (such as effective diffusion and catalytic rate coefficients of aggrecanases) need calibration and new experimental validation, as also recognized elsewhere [10]. Another drawback of the biochemical model is the exclusion of electric charge-based partitioning (Gibbs–Donnan effect) of IL-1 at the cartilage–medium interface. In fact, an updated version of a cytokine diffusion model [9] included the Nernst–Planck equations which would account for the electric charges of IL-1 (valency of -1 [9]) and Donnan partitioning. We are aware that the charge of cytokines may play a role in cytokine diffusion, but this was beyond the scope of the present study. We presume that Donnan partitioning associated with the negative charge of both IL-1 and GAGs may result in slightly smaller FCD loss if this effect was included in our model.

Another drawback is associated with modeling in 2D rather than 3D. The latter is especially important for future applications with patient-specific PTOA predictions, which in turn would need validation of model parameters in human instead of bovine cartilage. We chose to model degradation of bovine tissue in 2D to develop and investigate the computational concept of combined biochemical and biomechanical degradation mechanisms. At this step of model development, the concept was tested using a simple geometry in tandem with repeatable and relatively inexpensive experimental tissue degradation models. However, we acknowledge that the diffusion of cytokines in 3D cartilage geometries most likely differs from the 2D situation. Moreover, the complex 3D state of strain might also deviate from simple 2D case and thus have an impact on biomechanical damage. In addition, we chose not to include crack propagation since this would be challenging to validate experimentally.

Although chondrocyte apoptosis is appreciable in cartilage samples treated with catabolic cytokines [40] and in highly sheared/compressed regions [79,80], we chose not to include cell death explicitly in our present models. These processes would introduce new time and depth-dependent cell death parameters in need of validation. In the model of this study, decreasing cell concentration **(1)** decreases local aggrecan biosynthesis (thus, decreases FCD content) but in turn **(2)** leads to smaller amount of available IL-1 receptors ($C^*$ in Eq (2) decreases) and subsequently decreases expression of aggrecanases (thus, increases FCD content). The current model parameters should be critically scrutinized, since now in a preliminary model with decreased initial cell concentration the net effect of these competing processes increases FCD content compared to a model with unaltered amount of cells.

The biomechanical model without the base degeneration reached a near-equilibrium state in terms of FCD loss. We acknowledge that this might not be a real-life scenario; for example, FCD loss might lead to higher fluid velocity [49], altered shear strain threshold ($\hat{\varepsilon}_{threshold}$ in Eq (6)), or crack propagation during dynamic loading which in turn might lead to accelerated FCD loss. However, the model with the selected assumptions led to consistent results with the experiments.

In these mechanobiological models, we concentrated only on the non-fibrillar (*i.e.*, proteoglycan) matrix loss and purposely did not include the collagen degradation present in the original biochemical model [10]. Based on earlier animal model experiments [55], the loss of collagen is negligible over short periods of time post-injury. Also, in our preliminary models, we observed that inclusion of the collagen degradation did not have a substantial effect on tissue shear strains and thus on biomechanical degradation.

## Future

The functional significance of mechanobiological model predictions is yet to be fully appreciated. In the future, computer models like the ones used in the current study could facilitate prediction of early PTOA progression, recognition of high-risk chondral lesions, and simulation of treatment and rehabilitation outcomes. However, these aims need more experimental data on cartilage matrix loss at early times post-trauma via inflammatory mediators and shear strain-modulated tissue failure. Such data could also provide new information of possible "therapeutic window of opportunity" for effective drug interventions of PTOA [40,92]. Therefore, we plan to conduct new experiments with simultaneous biochemical and biomechanical degradations after injurious loading. This will be done at several timepoints to calibrate the degeneration rate parameters and to validate the mechanobiological algorithm.

The computational models should also be brought from tissue to joint level in order to predict PTOA progression for clinical relevance. This could be possible with synovial fluid or serum data on biochemical mediators [93,94] combined with subject-specific joint models

obtained via medical imaging [18]. For example, it would be interesting to include predictions of inflammation and biomechanically-driven degradation of ACL injured patients. Furthermore, besides joint trauma, increased levels of inflammatory mediators are suggested to play a role in the phenotype of osteoarthritis induced by obesity [95].

The current study also demonstrates the high need for fast and robust programming environments or software which could handle both inflammatory and biomechanical aspects simultaneously. This would reduce the strenuous coupling of several solvers and make the approach less complex and faster. Currently, the COMSOL Multiphysics can run the biochemical simulation alone in two minutes, but due to the complex biomechanical material model in ABAQUS the full simulation with both degradation mechanisms takes three days.

## Conclusions

Biochemical cues and abnormal mechanical environment are potential contributors linking knee trauma to progressive cartilage degeneration. Our numerical models considered these two cartilage-degrading mechanisms simultaneously for the first time in finite element models of mechanically injured cartilage, providing novel mechanobiological insights into PTOA progression. The model predictions were supported by current literature; cartilage subjected to pro-inflammatory stressors degraded near free surfaces, whereas in injured and dynamically loaded cartilage the FCD content was depleted especially near tissue lesions. Moreover, we suggest that the occurrence of chondral lesions **(1)** might affect the cytokine diffusion-driven degradation to some extent, and **(2)** could be the trigger for further accelerated biomechanical damage. In the future, mechanobiological models could evolve to *in silico* analysis tools that provide clinicians decision-making guidance for treatment of PTOA. This could lead to increased quality of life of patients via forestalling the onset of PTOA.

## Supporting information

**S1 Supplementary Material. Electronic supplementary material.** The supplementary material containing more detailed information about the underlying experimental findings, biochemical and biomechanical models, finite element mesh density, physiological levels of interleukin-1, and estimations of degraded cartilage area over time.
(DOCX)

**S1 Fig. Mesh sensitivity analysis.** Average fixed charge density (FCD) loss at time $t$ = 4 d with combined biochemical and biomechanical degradation with A) 499, B) 697, C) 918, D) 1217, and E) 2029 elements. The mesh with 918 elements was chosen, as increasing the mesh density from this did not yield quantitatively nor qualitatively different predictions for average FCD loss anymore.
(TIF)

**S2 Fig. Experimental findings in the absence of cytokine interleukin-1 suggest that cartilage loses more of its fixed charge density content near lesions compared to areas away from lesions.** Quantification of average optical density (OD) and fixed charge density (FCD) loss in freely-swollen control samples, uninjured dynamically loaded samples and injured dynamically loaded samples treated for 7 or 12 days. These biomechanical degradation experiments were carried out by Orozco et al. [49]. In injured samples, ODs were calculated as an average within 0.1 mm (±10%) from a lesion. In regions away from lesions, ODs were calculated as an average from a 0.45 mm (±10%) wide and 15 mm (±10%) thick surface region at the midway between lesion and sample edge. In freely-swollen and uninjured dynamically loaded samples, ODs were calculated as an average from a 0.45 mm (±10%) wide and 15 mm

($\pm$10%) thick surface region at the middle of samples. A) At day 0, average ODs between the freely-swollen and injured samples were similar (one-way ANOVA, $p = 0.068$). B) Average OD decreases significantly over time both away from lesions (one-way ANOVA, $p = 5.6 \cdot 10^{-4}$) and near lesions ($p = 2.6 \cdot 10^{-7}$; the figure shows Tukey's honestly significant difference (HSD) test results). C) At days 0, 7 and 12, average ODs were statistically similar between freely-swollen, uninjured dynamically loaded, and away from the lesion -groups (one-way ANOVA, $p = 0.105$ for day 0, $p = 0.213$ for day 7, $p = 0.416$ for day 12). However, within these treatment groups the average ODs decreased in time, especially in the away from the lesion -group which exhibited statistically significant decrease in OD (one-way ANOVA, $p = 0.065$ for freely-swollen samples, $p = 0.075$ for uninjured dynamically loaded samples, and $p = 5.6 \cdot 10^{-4}$ for injured dynamically loaded samples away from lesions; the figure shows Tukey's HSD test results). D) FCD losses (calculated from average ODs at day 7 and 12 compared to day 0) near lesions were significantly greater than away from lesions (the figure shows dependent samples *t*-test results). Box plots display values as range (brackets), interquartiles and median (solid bars). (TIF)

**S3 Fig. Average cartilage degradation in the whole explant geometry.** Simulated average bulk fixed charge density (FCD) losses in the whole explant with biochemical, biomechanical (with and without base degeneration, see S1 Supplementary Material Subsection S1.8) and combined biochemical and biomechanical degradation models. (TIF)

**S4 Fig. Cartilage degradation with physiological concentrations of interleukin-1.** A) Predicted biochemically driven fixed charge density (FCD) losses A) over time under moderate inflammation (1 ng/ml of exogenous interleukin-1) and physiological levels (0.01 ng/ml) of pro-inflammatory mediators. Still images at day $t = 4$ d show B) markedly higher matrix losses with moderate inflammation compared to C) physiological conditions. (TIF)

**S5 Fig. Still image preview of animations of cartilage degradation.** Animations of predicted fixed charge density (FCD) losses over 21 days of biochemical, biomechanical (without or with base degeneration), and combined biochemical and biomechanical degradation. (TIF)

**S1 Table. Compositional and material parameters for the fibril-reinforced porohyperelastic swelling (FRPHES) material model.** $z$ is the normalized distance from the cartilage surface ($z = 0$) to the bottom ($z = 1$). (PNG)

**S1 Animation. Animation of simulated biochemical degradation.** (GIF)

**S2 Animation. Animation of simulated biomechanical degradation without base degeneration.** (GIF)

**S3 Animation. Animation of simulated biomechanical degradation with base degeneration.** (GIF)

**S4 Animation. Animation of simulated combined biochemical and biomechanical degradation.** (GIF)

## Acknowledgments

The authors appreciate the support of the University of Eastern Finland and the Massachusetts Institute of Technology to conduct this study. CSC–IT center for Science Ltd., Finland, is acknowledged for providing the modeling software. Patrick Grahn, D.Sc. (Tech.), from COMSOL Multiphysics Support is acknowledged for technical support.

## Author Contributions

**Conceptualization:** Atte S. A. Eskelinen, Petri Tanska, Cristina Florea, Gustavo A. Orozco, Petro Julkunen, Alan J. Grodzinsky, Rami K. Korhonen.

**Data curation:** Atte S. A. Eskelinen, Cristina Florea, Gustavo A. Orozco.

**Formal analysis:** Atte S. A. Eskelinen, Cristina Florea.

**Funding acquisition:** Petri Tanska, Petro Julkunen, Rami K. Korhonen.

**Investigation:** Atte S. A. Eskelinen, Petri Tanska, Cristina Florea, Gustavo A. Orozco, Rami K. Korhonen.

**Methodology:** Atte S. A. Eskelinen, Petri Tanska, Gustavo A. Orozco, Petro Julkunen, Alan J. Grodzinsky, Rami K. Korhonen.

**Project administration:** Petri Tanska, Rami K. Korhonen.

**Resources:** Petri Tanska, Alan J. Grodzinsky, Rami K. Korhonen.

**Software:** Atte S. A. Eskelinen, Petri Tanska, Gustavo A. Orozco.

**Supervision:** Petri Tanska, Petro Julkunen, Alan J. Grodzinsky, Rami K. Korhonen.

**Validation:** Atte S. A. Eskelinen.

**Visualization:** Atte S. A. Eskelinen.

**Writing – original draft:** Atte S. A. Eskelinen.

**Writing – review & editing:** Petri Tanska, Cristina Florea, Gustavo A. Orozco, Petro Julkunen, Alan J. Grodzinsky, Rami K. Korhonen.

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
