## [Decision Letter · Decision Letter 0]

16 Apr 2020

Dear Mr. Eskelinen,

Thank you very much for submitting your manuscript "Mechanobiological model for simulation of injured cartilage degradation via pro-inflammatory cytokines and mechanical stimulus" for consideration at PLOS Computational Biology. As with all papers reviewed by the journal, your manuscript was reviewed by members of the editorial board and by several independent reviewers. The reviewers appreciated the attention to an important topic. Based on the reviews, we are likely to accept this manuscript for publication, providing that you modify the manuscript according to the review recommendations.

Sincerely,

Alison Marsden

Associate Editor

PLOS Computational Biology

Daniel Beard

Deputy Editor

PLOS Computational Biology

[LINK]

Reviewer's Responses to Questions

**Comments to the Authors:**

Reviewer #1: This paper presents a new finite element model to predict the simultaneous effect of degrading mechanisms on FCD content. This model incorporates diffusion of pro-inflammatory cytokine interleukin-1 into tissue and the effect of excessive levels of shear strain near chondral defects during physiologically relevant loading. Comsol, Abaqus and Matlab are used together to implement this complicated computational model. Good agreements with experiments and literature results are obtained.

Reviewer #2: Summary: Eskelinen et al. describe a novel finite element model to simulate spatial and temporal changes of FCD content in injured cartilage. The model incorporates (1) diffusion of the proinflammatory cytokine interleukin-1 into tissue, and (2) the effect of excessive levels of shear strain near chondral defects. They suggest that the presence of cartilage lesions plays a role in cytokine diffusion-driven degradation, and predisposes cartilage for further biomechanical degradation.

Osteoarthritis (OA) is an important disease which affects many people and a complete therapy still remains elusive. Therefore, further basic and translational research studies focusing on underlying mechanism of pathogenesis and potential diagnostic strategies are needed. Thus, the manuscript would contribute towards solving an unmet clinical need.

The manuscript is very well and comprehensively written including copious and detailed additional information and rationales in the supplementary. The abstract is straightforward and adequate. The introduction includes all needed information and gives an overview on the disease and pathogenesis as well as the clinical need. The M&M and the results section are clearly and understandable written.

The discussion is critical, all important limitations have been mentioned, and includes extensive literature search.

Specific Comments

Introduction

line 76 "on the other hand" makes no sense here

line 78/79 cartilage ECM - better to say ECM of cartilage

line 79/80 Cytokines originate primarily from cells in synovial lining, and they can recruit more cytokines to the inflamed area – wrong wording – additional cells/immune cells are recruited to the area + produce more cytokines - cytokines itself can’t be recruited

line 81 to specific chondrocyte (cartilage cell) receptors – better to say: cell receptors on chondrocyte surface

line 101 what does tissue stresses means?

Material and methods

Paragraph 2.2

The authors have implemented the Kar model in COMSOL Multiphysics, this code enables the simultaneous solution of PDEs and ODEs (which both occurs in the Kar model). The biomechanical part, i.e. the model of Orozco et al. has been modelled with ABAQUS. Thus, the main idea of the paper is to combine both models. As described also in the paper of Orozco the FCD loss has been computed using an iterative scheme implemented in MATLAB. Even after reading the original paper of Orozco it remains unclear whether this is done for each ABAQUS (fixed) grid point separately and how the number of iterations corresponds to the number of 21 days (as noted in line 238). Furthermore, it should be mentioned that the Kar model is solved by a time integration, whereby the biomechanical model solves a stationary problem (if the reviewer has understood the whole methodology correctly).

The variable MMP in the Kar model is obviously identical to ADAMTS in the manuscript, this should be mentioned and explained why it is here named differently.

In paragraph 2.2.2 some parts of the Kar model are presented using indices j1 and j2. Using j as general index and indices 1 and 2 as specific indices would be certainly clearer. If there are differences to the published model of Kar they should be mentioned in each case or the whole set of equations should be presented, but only in the supplement.

Most readers of the manuscript will have severe difficulties to understand paragraphs 2.2.6 to 2.2.8, to be honest also the reviewer. Perhaps it could be a good idea, to completely move the detailed description of the mechanical model (which can be similarly found in the paper of Orozco) into the supplement and to reduce the content of these paragraphs to a clear description of the input and the output of the approach without expecting detailed knowledge in elastomechanics.

The authors could also decide to give only a very short overview of the methodology, like it was done for the Kar model. Both would significantly shorten the length of the manuscript, even if the already extensive supplement could then be further extended.

Discussion

In the discussion all results of the simulations were carefully and probably exhaustively compared to a number of experimental findings. Also, several limitations of the approach were discussed in detail. For example, most of the large number of parameters within the Kar model are only "calibrated" but not validated by a parameter optimization including a sensitivity analysis. Thus, the manuscript shows, as promised in the abstract, that a simultaneous simulation of mechanical and biochemical processes is possible, even if there is still a long way to a validated full model for the processes during PTOA or, as a further example, rheumatic arthritis. There is also a strong demand for programming environments which can simultaneously perform simulations for interconnected biochemical and mechanical (and other) processes, without the tedious coupling of separate solvers for parts of the whole system of equations. To demonstrate this difficulty, the reviewer would recommend to present some examples for the computing times.

For the reviewer and probably also for prospective readers a list of abbreviations would be extremely helpful.

**Have all data underlying the figures and results presented in the manuscript been provided?**

Reviewer #1: Yes

Reviewer #2: Yes

PLOS authors have the option to publish the peer review history of their article (what does this mean?). If published, this will include your full peer review and any attached files.

Reviewer #1: No

Reviewer #2: No
---

## [Decision Letter · Decision Letter 1]

28 May 2020

Dear Mr. Eskelinen,

We are pleased to inform you that your manuscript 'Mechanobiological model for simulation of injured cartilage degradation via pro-inflammatory cytokines and mechanical stimulus' has been provisionally accepted for publication in PLOS Computational Biology.

Best regards,

Alison Marsden

Associate Editor

PLOS Computational Biology

Daniel Beard

Deputy Editor

PLOS Computational Biology

Reviewer's Responses to Questions

**Comments to the Authors:**

Reviewer #1: The paper is ready for acceptance.

Reviewer #2: The authors have carefully considered all comments and suggested changes by the reviewer, many thanks. The manuscript may be now accepted in the revised form.

**Have all data underlying the figures and results presented in the manuscript been provided?**

Reviewer #1: Yes

Reviewer #2: Yes

PLOS authors have the option to publish the peer review history of their article (what does this mean?). If published, this will include your full peer review and any attached files.

Reviewer #1: No

Reviewer #2: No

---

## [Editor Report · Acceptance letter]

16 Jun 2020

PCOMPBIOL-D-20-00152R1 

Mechanobiological model for simulation of injured cartilage degradation via pro-inflammatory cytokines and mechanical stimulus

Dear Dr Eskelinen,

I am pleased to inform you that your manuscript has been formally accepted for publication in PLOS Computational Biology. Your manuscript is now with our production department and you will be notified of the publication date in due course.

With kind regards,

Sarah Hammond
